# Freeze-Thaw Effect on Riverbank Stability

**Chao Li [1], Zhen Yang [1],*, Hung Tao Shen [2] and Xianyou Mou [1]**

1   College of Water Conservancy and Civil Engineering, Inner Mongolia Agricultural University, Hohhot 010018, China

2   Department of Civil and Environmental Engineering, Clarkson University, Potsdam, NY 13699, USA

*   Correspondence: yangzhen@emails.imau.edu.cn

**Abstract:** The stability of riverbanks in cold regions is affected by the freeze-thaw action. The freeze-thaw process causes changes in the moisture content, friction angle, and cohesiveness of the bank material. Together with the freeze-thaw effect, seepage pressure influenced by the changing water levels, and the bank slope are the key factors affecting bank stability. A limit equilibrium bank stability model considering the infiltration water pressure effect is developed and applied to the Shisifenzi section of Inner Mongolia reach of the Yellow River. Laboratory tests of field samples with moisture contents of 15%, 18%, 21%, and saturated showed that the freeze-thaw action reduced the degree of saturation by 34.37 %, 30.71%, 32.48%, and 46.23%, respectively, accompanied by reductions in the internal friction angles by 1.78%, 2.74%, 6.33%, and 5.32%. These changes resulted in a 24.35% to 29.13% reduction in the safety factor of bank stability. Together with seasonal variations in the water levels the field data showed that the bank stability safety factor in the study site increases gradually through the melting period, dry period, wet period, flooding period, and low flow period. The slope stability safety factor increases with the stage of the river but decreases with the groundwater level.

**Keywords:** seasonal freezing zone; limit equilibrium; freeze-thaw erosion; pore water pressure; bank stability

## 1. Introduction

Bank stability is an important phenomenon affecting river channel morphology. For the Inner Mongolia reach of the Yellow River, the stability of riverbanks is essential for flood control and bank protection projects. Bank erosion also contributes to sediment supply to the river. The bank stability is mainly influenced by channel flow and the composition and properties of the riverbank material [1–5]. Erosion at the bank toe could lead to bank instability. Under the action of gravity, cracks will develop along the banks, especially when there is a significant difference between the groundwater level and water level in the channel due to the difference in the infiltration water pressures [6]. These cracks could be further enhanced by heavy rainfall and freeze-thaw action [7].

Bank stability is influenced by numerous factors [8,9]. It is mainly affected by the channel flow condition, embankment groundwater level, and sediment transport condition [3,10–12]. They are mostly manifested in the scouring and seepage effects, resulting in gravity failure of the bank [13,14]. The composition and properties of riparian soils are internal factors affecting the stability of riverbanks [4,15–19]. The type of bank instability may vary with the channel and cross-sectional profiles and bank material properties [20–22]. Regarding the slope and material properties, the more significant the proportion of cohesive components in riparian soils, the stronger their ability to withstand scouring [18]. However, soil cohesion and shear strength are affected by water content and freeze-thaw cycling [4,15,17,23]. The former affects the pore water pressure and matrix suction within the soil, thereby changing the scour resistance of the bank [24]. The latter alters the soil's mechanical properties, i.e., cohesion and internal friction angle, by weakening its internal structure [23,25].

Bank instability is often the result of multiple factors, and the pattern of bank instability can differ depending on the bank and hydraulic conditions [4,11,26–28]. Osman and Thorne [11] proposed the classical steep slope plane instability model for homogeneous clayey soil banks, which considers the influence of scouring on the bank boundary. Darby and Thorne [26] improved the Osman and Thorne [11] bank stability model by considering the effects of hydrostatic and pore water pressure in the force analysis of the damaged surface. In addition, other influencing factors, e.g., matrix suction, groundwater level, soil properties, etc., and dynamic water pressure have been analyzed [15,17,29–31]. For complex non-homogeneous multilayered soil banks, Amiri–Tokaldany [32,33] proposed a bank stability model for non-single soil layers based on Dandy and Thorne [26]. Lai [34] coupled bank stability analysis with a two-dimensional streambed dynamics model to analyze and predict the collapse of multilayered clayey soil bank slopes. This study will examine the stability of seasonal-frost banks with coupled hydraulic effects of channel water level and the groundwater level in the Shisifenzi Bend section of the Inner Mongolia reach of the Yellow River, as shown in Figure 1.

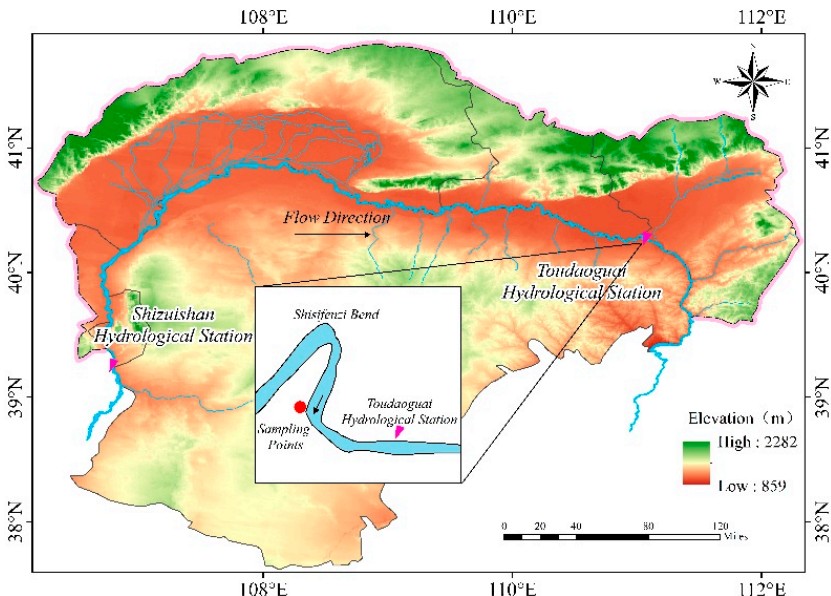

**Figure 1.** Inner Mongolia reach of the Yellow River, and the Shisifenzi Bend section.

The weather of the study area is controlled by the Mongolian high pressure and has a typical continental monsoon climate. The annual average temperature is 7.3 °C, the highest temperature is 38.4 °C and the lowest temperature reaches −36.3 °C. The average annual rainfall is 359.8 mm and the evaporation is 1849 mm. The freezing and thawing period is 4 months long, with the temperature turning negative in mid-November and positive in mid-March of the following year. The Inner Mongolia reach of the Yellow River has very gentle slopes. The bed slope is nearly 1/10,000, with many bends and broad, shallow cross-sections [35]. The riverbanks are mainly formed from the deposition of sediment from upstream and are prone to instability and collapse. Moreover, since it is situated in the seasonal frost zone, bank failures are more frequent due to the combined effect of hydraulic erosion and the freeze-thaw impact. This paper extends the Darby and Thorne bank stability model [26] by introducing the effect of infiltration water pressure. The model is applied to the Shisifenzi Bend for different river flow and groundwater conditions through the freeze-thaw period and other seasons of the year. Figure 2 shows an example of the bank failures in the Shisifenzi Bend section.

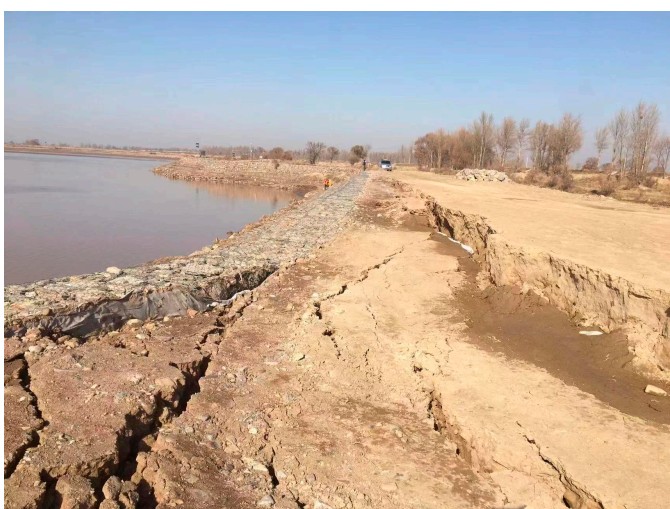

**Figure 2.** Yellow Riverbank at Shisifenzi, Inner Mongolia.

## 2. Slope Stability Analysis and Modeling

### 2.1. Bank Stability Mechanics

The critical height and limit equilibrium methods are commonly used in bank stability analysis [36]. The former applies to steep and vertical bank slopes, while the latter accounts for the actual geometric form of bank slopes in the stability analysis. The limit equilibrium method considers the stability safety factor, $F_S$, of the bank slope. When $F_S$ is below the critical value, it is assumed that the bank will be destabilized and collapse. The safety factor can be calculated as:

$$F_S = \frac{F_R}{F_D} \tag{1}$$

In which, $F_D$ is the sum of the component of all forces acting on the potential sliding mass along the sliding surface; $F_R$ is the resistance force on the potential sliding surface, which is a function of the cohesive force and frictional force. Xia [5] suggested that the bank remains stable when $F_S > 1.3$; and the bank is unstable when $F_S < 1.0$. When $1.0 \leq F_S \leq 1.3$, the bank is at risk of possible failure. The threshold of instability safety factor is set to 1.3 in the present study.

Forces on the Potential Sliding Mass

The bank is subjected to the gravity force, the lateral water pressure from the channel water and the pore water pressure from the bank. The pore water pressure is generated by the groundwater level inside the bank, which lags behind the change in channel water level. Figure 3 shows a force diagram of riverbank, in which W is the gravity force, $kN/m^2$; $F_{cp}$ is the lateral hydrostatic pressure, $kN/m^2$; $F_x$ and $F_z$ are the horizontal and vertical components of $F_{cp}$, respectively; $\omega$ is the angle between $F_{cp}$ and the horizontal direction, $°$; $\alpha$ is the angle between $F_{cp}$ and the potential sliding surface, $°$; $U_w$ is the pore water pressure, $kN/m^2$; $P_d$ is the infiltration water pressure, $kN/m^2$; $e$ is the angle between the infiltration water pressure and the normal direction of the sliding surface, $°$; $h_w$ is the groundwater level, m; $\xi$ is the river water level, m; $\beta$ is the inclination of the potential sliding surface, $°$; $i$ is the slope inclination, $°$; $b$ is the top width of the potential sliding mass, m; $H$ is the potential sliding depth, m; $H_1$ is the depth of the bank slope, m; $H_2$ is the depth of the bank tensile fracture, m; $H_3$ is the depth of the residual tensile fracture, m.

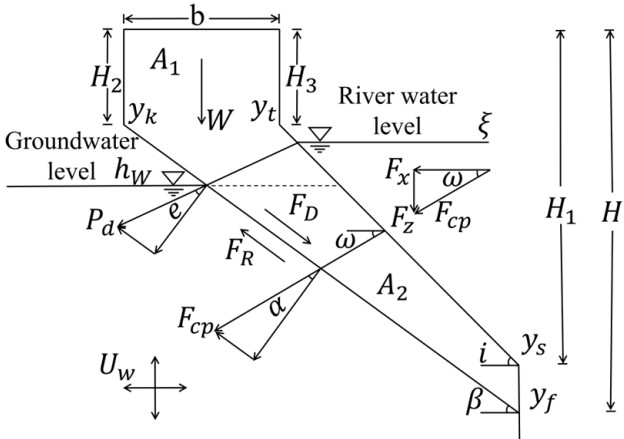

**Figure 3.** Schematic diagram of forces on the sliding surface of the riverbank (after Darby and Thorne with modification.).

*Gravity*: Based on the channel water level and the groundwater level elevations, the sliding mass is divided into two parts by the groundwater table: above it is the unsaturated soil, area $A_1$, m$^2$; below it is the saturated soil, area $A_2$, m$^2$. The gravity is calculated as:

$$W = \gamma A_1 + \gamma_{sat} A_2 \tag{2}$$

In which, $\gamma$ is the specific weight of the soil, kN/m$^3$; $\gamma_{sat}$ is the specific weight of saturated soil, kN/m$^3$. and

$$A_1 = \frac{1}{2}\left[\left(-H_2^2 + 2Hh_w - h_w^2\right)\tan\beta + \left(H_3^2 + 2H_1h_w + h_w^2\right)\tan i\right] A_2 = \frac{1}{2}[(H - h_w)^2\tan\beta - (H_1 - h_w)^2\tan i] \tag{3}$$

*Lateral hydrostatic pressure*: The hydrostatic pressure under the water surface is $P = \gamma_w h$, where, $P$ is the hydrostatic pressure, kN/m$^2$; $\gamma_w$ is the specific weight water and is set to 10 kN/m$^3$ consider the turbidity of the river water; $h$ is the depth from the water surface, m. The hydrostatic pressure can be decomposed into horizontal hydrostatic pressure $F_x$ and vertical hydrostatic pressure $F_z$, then the total hydrostatic pressure is:

$$F_{cp} = \sqrt{F_x^2 + F_z^2} \tag{4}$$

$F_x$ and $F_z$ are calculated by:

$$F_x = \frac{1}{2}\gamma_w h^2 \tag{5}$$

$$F_z = \gamma_w S \tag{6}$$

$$\omega = \tan^{-1}\frac{F_z}{F_x} \tag{7}$$

where $S$ is the area of the water body above the sliding mass, m$^2$.

*Pore water pressure*: The pore water pressure at any point on the sliding surface is $U_w = \gamma_w(h_w + \xi)$. The total pore water pressure acting on the potential sliding body is:

$$U_w = \sum_0^x u_w dx \tag{8}$$

*Infiltration water pressure*: The infiltration water pressure can be calculated by:

$$P_d = \gamma_w J A_2 \tag{9}$$

where $J$ is the infiltration gradient, $J = \frac{\Delta h}{l}$; $l$ is the seepage path length, m; $\Delta h$ is the head difference, m.

*Sliding surface angle*: The sliding surface angle, $\beta$, of the collapsed body can be estimated by Equation (10) [37]:

$$\beta = \frac{i + \varphi}{2} \tag{10}$$

where $\varphi$ is the internal friction angle of the slope soil, $^\circ$.

### 2.2. Modeling Bank Slope Instability

Depending on the interaction between the channel water and the bank, the bank failure mode varies, including plane sliding, arc sliding, and slumping. Specifically, arc sliding failure occurs when the bank slope is gentle, plane sliding failure occurs when the bank slope is steep, and slumping collapse occurs when the clay layer above a binary structure bank is suspended by erosion below [4,27,38,39]. Since the bank slope of the study reach is steep, and the soil is mainly low liquid limit powder, the bank can be considered monolithic. The main purpose of this paper is to analyze the stability of the riverbank and to facilitate the calculation according to the plane sliding analysis. Osman and Thorne [11] divided riverbank collapse into initial and secondary collapse based on plane sliding. The initial failure is the scouring of the bank toe, and the secondary collapse refers to the parallel retreating process of the riverbank and the sliding surface passing through the bank toe. Based on this assumption, they proposed a model for calculating the stability of the riverbank in the case of plane failure. However, the model failed to consider lateral water pressure, pore water pressure, infiltration water pressure, and the damaged surface must pass through the slope foot. Darby and Thorne [26] improved the Osman and Thorne [11] model so that the sliding surface is not necessary to pass through the bank toe, and the effects of lateral water pressure and pore water pressure in the force analysis of the sliding surface are considered. The established stability calculation model for riverbanks with clayey soils is:

$$F_S = \frac{F_R}{F_D} = \frac{C'L + [(W - U_W)\cos\beta + F_{CP}\cos\alpha]\tan\varphi'}{W\sin\beta - F_{CP}\sin\alpha} \tag{11}$$

In which, $C'$ is the effective cohesive force of soil, kPa; $\varphi'$ is the effective internal friction angle of soil, $^\circ$.

Although the lateral and pore water pressures are considered in the model of Darby and Thorne [26], the model does not consider the role of infiltration water pressure. The infiltration water pressure acting on the sliding body is an important factor affecting the stability of the river embankment slope, especially during the freeze-thaw period when the channel water level declines and during the summer flood period when the river water level varies greatly.

Since the sliding force $F_D$ is a combination of the gravitational force, hydrostatic pressure, and infiltration water pressure acting on the potential sliding body along the direction of the sliding surface. Equation (12) modifies Darby and Thorne [26] formulation by including the infiltration water pressure in the sliding force calculation:

$$F_D = W\sin\beta - F_{CP}\sin\alpha \pm P_d\sin e \tag{12}$$

The infiltration water pressure is negative when the river water level is higher than the groundwater level and positive when the groundwater level is higher.

By taking into account the effects of gravity, pore water pressure, lateral hydrostatic pressure, and infiltration water pressure, the slip resistance of the potential sliding surface is given by:

$$F_R = C'L + [(W - U_W)\cos\beta + F_{CP}\cos\alpha \pm P_d\cos e]\tan\varphi' \tag{13}$$

Substituting Equations (12) and (13) into Equation (1) gives bank stability factor, $F_S$:

$$F_S = \frac{F_R}{F_D} = \frac{C'L + [(W - U_W)\cos\beta + F_{CP}\cos\alpha \pm P_d\cos e]\tan\varphi'}{W\sin\beta - F_{CP}\sin\alpha \pm P_d\sin e}, \tag{14}$$

## 3. Model Application and Analysis

### 3.1. Study Site

The modified bank stability model is used to analyze the bank stability in the Shisifenzi Bend of the Yellow River in Inner Mongolia, located at 40°17′39″ N and 111°2′53″ E in Tuoketuo County, 4 km upstream of the Toudaoguai hydrological station. as shown in Figure 1. The river channel turns sharply from northwest to southeast. The channel slope is about 0.1%. The channel width varies between 200 m and 500 m.

Field samples indicate the bank material mainly consists of sandy loam and shallow clay. Table 1 lists the basic physical and mechanical parameters of the soils obtained by laboratory tests.

**Table 1.** Basic physical and mechanical parameters of bank material.

| Natural Specific Weight, (kN/m³) | Saturated Specific Weight, (kN/m³) | Median Particle Size, $D_{50}$/(μm) | Dry Density, (g/cm³) | Experimental Values | | Valid Values | |
|---|---|---|---|---|---|---|---|
| | | | | $C$, (kPa) | $\varphi$ | $C'$, (kPa) | $\varphi'$ |
| 18.1 | 19.3 | 33.55 | 1.58 | 35.5 | 30 | 24.85 | 27 |

Note: The effective cohesive force $C'$ in the Table is 0.7 times the experimental value; the effective internal friction angle $\varphi'$ is 0.9 times the experimental value [40].

### 3.2. Bank Stability Analysis

The slope of the riverbank in the study area varies between 30° and 60°. An average value of 45° is adopted for calculation. Figure 4 shows the water level in the study reach changes rapidly over a wide range. The lowest water level was 987.14 m on May 7, while the highest was 990.65 m on September 6, with a variation of 3.51 m. The top elevation of the bank is about 992.0 m, and the height is 8 m. Hence the riverbank is modeled with $H = 8$ m, $H_1 = 7$ m, $H_2 = H_3 = 2$ m, and $i = 45^o$, as shown in Figure 5. Seven bank groundwater levels varying from 0 m to 3.0 m are used to calculate the safety coefficient of bank stability, with the river water level ranging from 1 to 8 m from the top of the bank under each working condition.

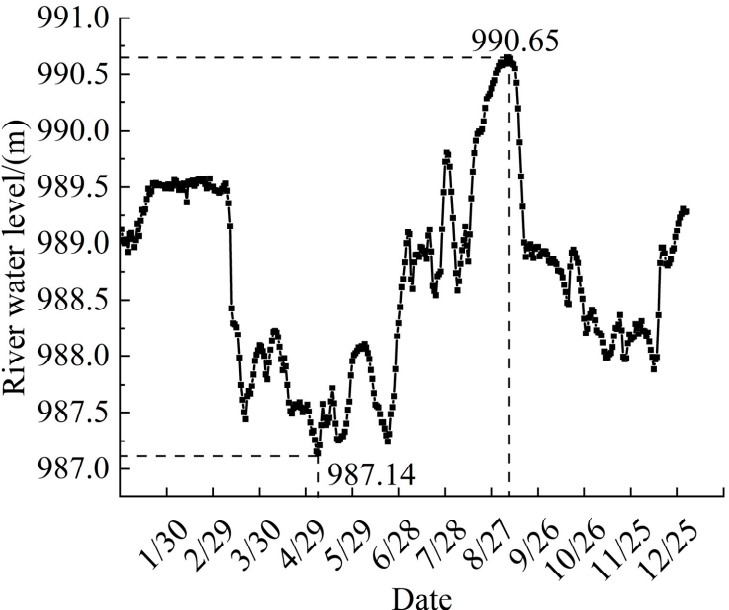

**Figure 4.** Daily average water level at the Shisifenzi in 2020.

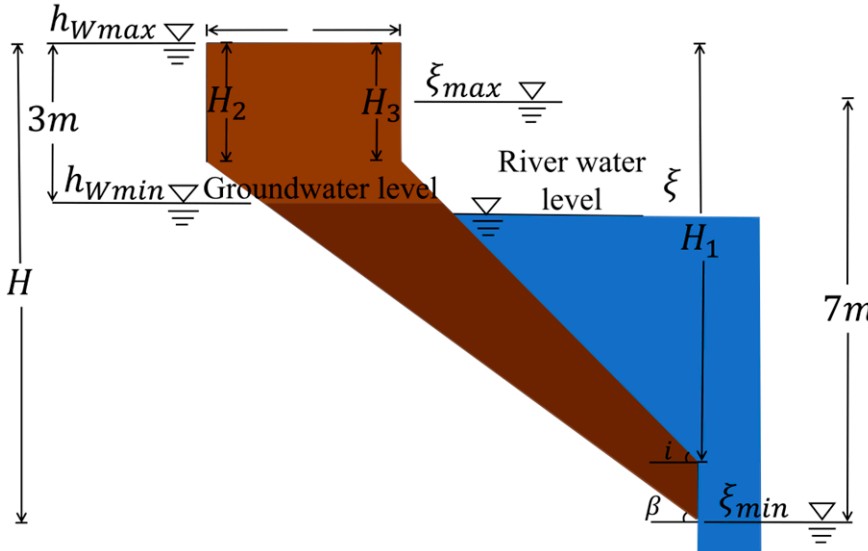

**Figure 5.** Calculated bank slope profile.

Figure 6 shows the variation of bank stability factors under different combinations of groundwater and river water levels. Table 2 shows the stability of the riverbank under different combinations of conditions and the critical stability safety factor of 1.3. From the calculated results shown in Figure 6 and Table 2, it can be seen that: the stability safety factor of the bank slope decreases rapidly with the lowering of the river water level, and finally tends to be stable. When the stability safety factor of bank slope is less than 1.3, the bank slope will be destabilized. The rise and fall of river water level will cause the change in groundwater level in the bank, which makes the water content, the cohesion, and the internal friction angle of the bank soil change.

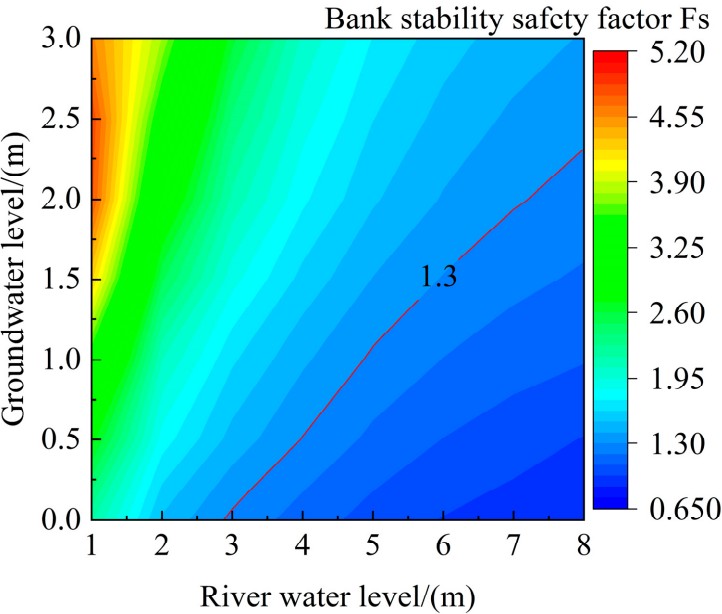

**Figure 6.** Variation of bank stability factor with groundwater levels and river water levels.

**Table 2.** Bank slope stability under different combinations of groundwater level and river water level.

| Groundwater Level $h_w$/m<br>River Water Level $\zeta$/m | 0 | −0.5 | −1 | −1.5 | −2.0 | −2.5 | −3.0 |
|---|---|---|---|---|---|---|---|
| −1 | + | + | + | + | + | + | + |
| −2 | + | + | + | + | + | + | + |
| −3 | - | + | + | + | + | + | + |
| −4 | - | - | + | + | + | + | + |
| −5 | - | - | - | + | + | + | + |
| −6 | - | - | - | - | + | + | + |
| −7 | - | - | - | - | + | + | + |
| −8 | - | - | - | - | - | + | + |

Note: + and - indicates stable and unstable conditions.

The reach studied has a freezing period of nearly four months, during which the river water level changes significantly under the influence of ice conditions. In the meantime, the bank freeze and thaw affect the mechanical properties of the bank material. To determine the seasonal variation of the slope stability, according to the changes in temperature and river water level, the flow period is divided into the thawing period (early March–end of March), dry period (early April–end of June), wet period (early July–early September), flooding period (mid–September), and low flow period (late September–mid-November). The safety coefficient of the bank stability in each period is calculated. The calculated result is presented in Table 3.

**Table 3.** Bank stability factors for different periods of the year.

| Parameters | Thawing Period | Dry Period | Wet Period | Flooding Period | Low Flow Period |
|---|---|---|---|---|---|
| Cohesion $C'$/kPa | 16.8 | 16.8 | 24.85 | 24.85 | 24.85 |
| Friction angle $\varphi'$/° | 25.2 | 25.2 | 27 | 27 | 27 |
| Ground water level $h_w$/m | 0 | −3 | −3 | −0.5 | −0.5 |
| River water level $\zeta$/m | −5 | −5 | −1.5 | −1.5 | −4 |
| Safety factor | 0.75 | 1.26 | 4.16 | 2.14 | 1.29 |

As shown in Table 3, the safety factor increases from the thawing period to the high-water periods and decreases during the receding water period. The safety coefficient during the thawing period is only 0.75, far smaller than the threshold of 1.3, indicating a period of potential destabilization of the bank slope. The main reasons are: (1) After the freeze-thaw cycle, the cohesion and internal friction angle of the bank material is reduced, and the shear strength is weakened; (2) During this period, the river water level drops sharply after the melting of ice cover. The lateral water pressure decreases, while the infiltration water pressure towards the river rises. Under the combined action of these two, the bank slope is destabilized; (3) During the dry period, the soil is still impacted by the freeze-thaw effect. However, the difference between the groundwater and river water levels decreases, so the infiltration water pressure effect is weakened compared to the thawing period. Hence, the safety coefficient of the bank increased, but it is still lower than the 1.3 threshold, and the bank slope is still subjected to destabilization; (4) When the river water level rises during the wet water period, the lateral hydrostatic pressure increases. Although the change in groundwater level in the bank lags behind, and there is infiltration water pressure towards the bank. Hence, the safety factor is the highest; (5) During the flooding period, the groundwater level of the bank rises and is slightly higher than the river water level. The infiltration water pressure is small, and the safety factor is smaller than the wet water period. However, the bank slope is stable in this period; (6) The river water level and the lateral water pressure drop during the low water period. However, the groundwater level of the bank is still high, and there is a large infiltration water pressure toward the river. In this case, the safety factor of the bank slope is reduced to 1.29. These indicate that the

bank slope may still be susceptible to instability during the low water period, which partly explains why the sand content of the river flow is still high in September and October after the flooding period.

Through the statistical analysis of the monthly average sediment content of the River at the Shizuishan and Toudaoguai hydrological stations from 2010 to 2019, as shown in Figure 7, it can be seen that the sand content of the Shizuishan section is higher than that of the Toudaoguai section during the freezing period (December–February). This means that the river channel is subjected to siltation during the freezing period. However, the sediment content of the Toudaoguai section is significantly higher during the open water period (March–November). In this period, the sediment supply is mainly from tributaries, i.e., the Ten Kungdui, Kundulun River, Wudangou, etc., and the collapsed bank material. During the summer flood period (June–September), the sand is mainly from the tributaries during flash floods. The spring flood period (March–April) is in the thawing and dry period, and the stability and safety coefficient of the bank slope is the smallest. The sediment mainly comes from the bank collapse under the effect of freeze-thaw erosion. From Figure 7, it can be seen that the average sediment content of the Toudaoguai section reaches 2.5 kg/m$^3$ in March–April, the highest sediment content period second only to the summer flood season. Therefore, it can be concluded that the effect of freezing and thawing on the collapse and destabilization of the bank slope and on the sediment content in the river channel cannot be ignored.

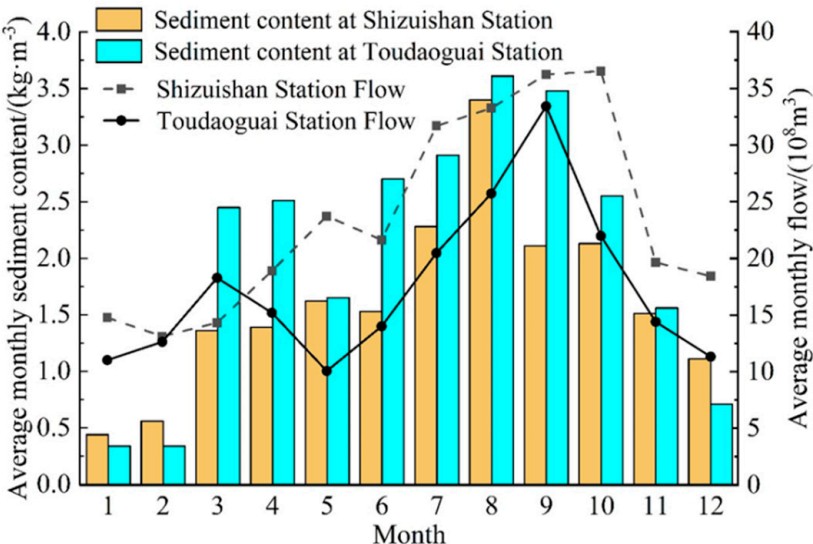

**Figure 7.** Monthly average sediment content from 2010–2019.

### 3.3. Factors Affecting Bank Stability

The bank stability is governed by hydraulic and bank conditions and the bank condition. The hydraulic condition is mainly caused by the change in water level inside and outside the river channel. The bank condition includes the bank slope, soil composition, and physical and mechanical properties. The river water level changes a lot during the flood and summer flood periods, but the groundwater level of the bank lags behind, which produces the infiltration water pressure on the bank. Moreover, the freezing and thawing effect of the flood will change the soil properties. The effects of freeze-thaw action, infiltration water pressure, and bank slope on bank stability are discussed in this section.

#### 3.3.1. Freeze-Thaw Effect

The Shisifenzi Bend has long, cold winters, with the air temperature remaining below freezing from mid-November to mid-March. The freeze-thaw period lasts for more than four months. The frozen zone surface will turn into a water-rich layer when it melts [41]. Hence, the groundwater level is set at 0 m at the end of the freeze-thaw period. The stability

of the embankment soil before and after the freezing and thawing is calculated. Remodeled field soil samples with water contents of 15%, 18%, 21%, and fully saturated samples are used to investigate the effect of freeze-thaw action on their mechanical properties. To fully freeze-thaw the samples, they were frozen at −15 °C for 12 h then thawed at 15 °C for another 12 h. The number of freeze-thaw cycles was selected as 0, 1, 3, 5, 7, 9, 11, 13, and 15. The changes in cohesion and internal friction angle of the samples with varying moisture contents after different freeze-thaw cycles were obtained using the static triaxle test with and Mohr–Coulomb failure criterion, as shown in Figure 8. The cohesion c and internal friction angle φ of soil samples with each moisture content became stable after 15 freeze-thaw cycles. Figure 8a shows that the cohesion c of each moisture content soil sample tends to level off after 15 cycles and decreased by 34.37%, 30.71%, 32.48%, and 46.23%, respectively. Correspondingly, the internal friction angle φ declined by 1.78%, 2.74%, 6.33%, and 5.32%, respectively, as shown in Figure 8b.

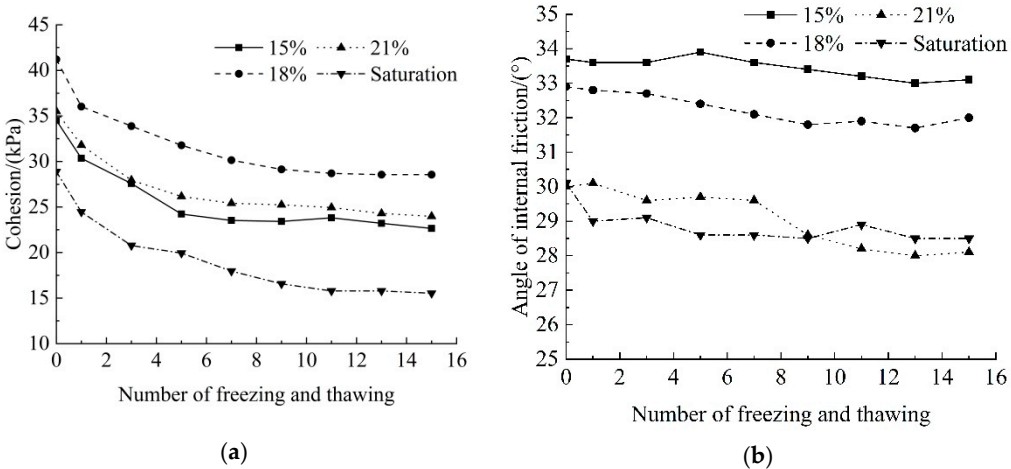

**Figure 8.** Effect of freezing and thawing on soil cohesion and angle of internal friction. (**a**) Soil cohesion, (**b**) Angle of internal friction.

The water content of the undisturbed soil sample measured on site is about 21%. The samples were taken on 15 July 2020, when the river water was rising. The effective cohesion and effective internal friction angle of soil samples with 21% moisture content after 0 and 15 freeze-thaw cycles are used to calculate the bank stability safety factor before and after freeze-thaw, as shown in Figure 9. It can be seen that the stability safety factor of the bank after freeze-thaw is significantly lower.

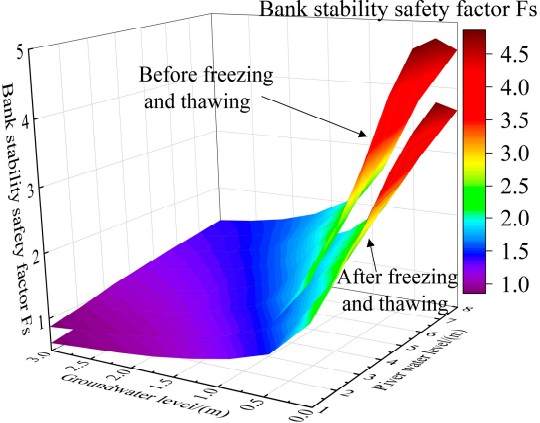

**Figure 9.** Effect of freezing and thawing on bank stability.

### 3.3.2. Effect of Infiltration Water Pressure

To study the effect of the infiltration water pressure, the safety factor of the bank stability with or without considering the infiltration water pressure when the groundwater level is 1 m, 2 m, and 3 m, and the river water level varies from 1 to 8 m from the top of the bank is calculated. The result is shown in Figure 10. When the groundwater level is lower than the river level, the infiltration water pressure is toward the bank, which inhibits the sliding of the slope. The bank stability safety factor is larger than when the infiltration water pressure is not considered. On the other hand, when the groundwater level is above the river water level, the infiltration water pressure is towards the river. As a result, the sliding force on the sliding mass increases, which promotes the destabilization of the slope. Therefore, the safety factor when infiltration water pressure is considered is smaller.

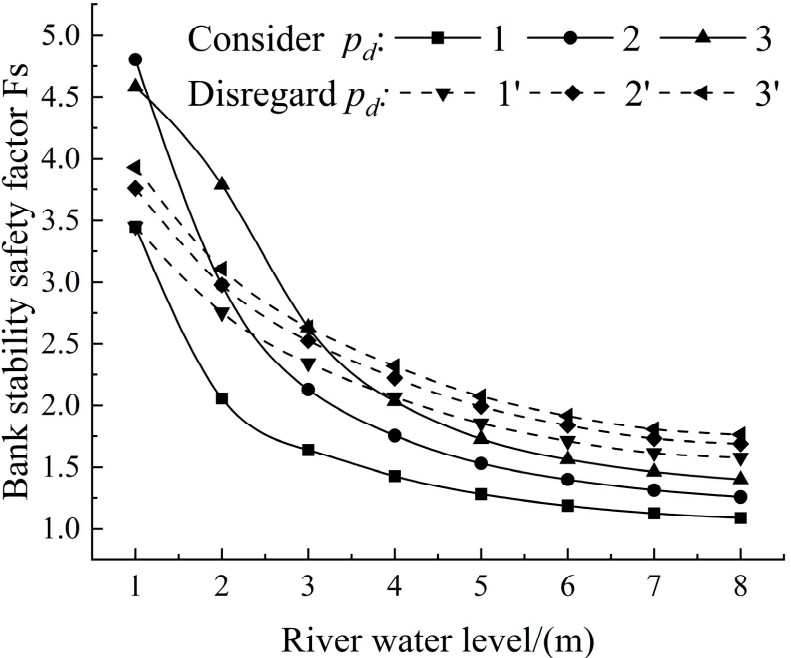

**Figure 10.** Effect of infiltration water pressure on bank stability.

### 3.3.3. Effect of Bank Slope

To examine the effect of bank slope, the bank stability factor is calculated for the slope angles of 30°, 35°, 40°, 45°, 50°, 55°, and 60°, respectively, and the groundwater level is 1 m. As shown in Figure 11, the bank stability factor varies significantly with the bank slope. When the river water level is lower than the groundwater level, the infiltration water pressure is generated, which promotes the destabilization of the bank slope. When the river water level is below 4 m, there is a possibility of bank collapse under different bank slope angles. In addition, the safety factor of the bank decreases gradually as the bank slope increases.

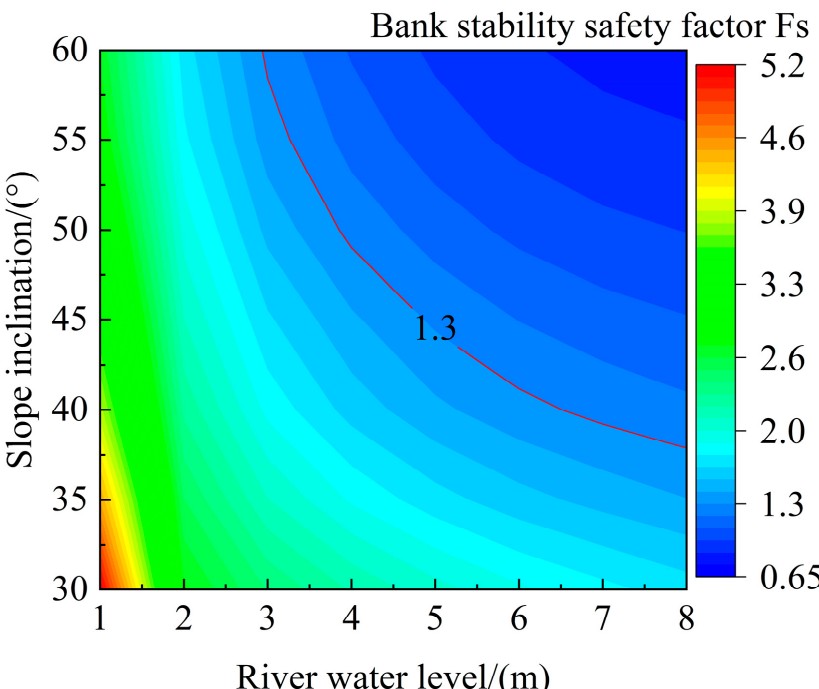

**Figure 11.** Effect of bank inclination on bank stability.

### 4. Conclusions

This paper extended the Darby and Thorne [26] bank stability model by considering the effect of infiltration water pressure, which is important for rivers subjected to freeze-thaw actions in the banks. The model is used to analyze the bank slope stability of the Shisifenzi Bend of the Yellow River in Inner Mongolia. The effects of freeze-thaw action, infiltration water pressure, and bank slope are studied. The major conclusions are:

(1) When the river water level is constant, the safety factor of bank stability decreases with the rising groundwater level. When the groundwater level is constant, the safety coefficient of bank stability declines with the decreasing river water level, with a trend of thawing period < dry period < low water period < flooding period < wet period.

(2) The freeze-thaw action significantly changes the mechanical properties of the bank material, leading to a 24.35–29.13% reduction in the safety factor of bank stability, indicating the important effect of the freeze-thaw in reducing the bank stability.

(3) When the groundwater level is lower than the river water level, the infiltration water pressure mainly shows an inhibitory effect on the bank stability. The safety factor of bank stability will increase. By contrast, when the groundwater level is higher, the infiltration water pressure plays a destabilizing role.

(4) When the river water level is below 4 m, there is a possibility of bank collapse under different bank slope angles, and the safety factor decreases with larger bank slope angles.

**Author Contributions:** Conceptualization, C.L. and Z.Y.; consultation, H.T.S.; methodology, C.L. and Z.Y.; model development, Z.Y. and C.L.; field and laboratory experiments, C.L. and Z.Y.; writing, Z.Y., C.L. and H.T.S.; funding acquisition, C.L. and X.M. All authors have read and agreed to the published version of the manuscript.

**Funding:** This research was funded by [National Natural Science Fund of China] grant number [No.51969025], [Natural Science Foundation of Inner Mongolia Autonomous Region] grant number [No.2019MS05006] And [Key project of Inner Mongolia Natural Science Foundation] grant number [No.2022ZD08].

**Institutional Review Board Statement:** Not applicable.

**Informed Consent Statement:** Not applicable.

**Data Availability Statement:** The data presented in this study are available on request from the corresponding author or the first author.

**Acknowledgments:** Special thanks to Xinchuan Lu of the Yellow River Wanjiazhai Hydraulic Hub Co., Ltd. for providing the water level data of the Shisifenzi reach.

**Conflicts of Interest:** The authors declare that they have no conflict of interest.

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
