# Peer review of "Freeze-Thaw Effect on Riverbank Stability"

_water, doi:10.3390/w14162479_

Round 1

Reviewer 1 Report

Freeze-Thaw Effect on River Bank Stability

Summary: This article uses the limit equilibrium bank slope stability model to analyze the stability of the Inner Mongolia reach of the Yellow River in different hydrological periods. The main research purpose of this paper is to consider the effect of freeze-thaw cycles on the stability of bank slopes. What is more novel is that not only the effects of freeze-thaw cycles on the mechanical parameters of bank slope materials are considered, but also the groundwater levels and river water levels of river banks caused by freeze-thaw cycles. It is of innovative significance to bring the changes caused by freezing and thawing into the stability model to calculate its safety factor and analyze the stability. Secondly, the paper also introduces the seepage water pressure, extends the Darby and Thorne river bank stability model, and analyzes the effect of seepage water pressure on the stability of the bank slope by bringing the seepage water pressure into the stability analysis model.

comments:

In general, this article is reasonable in terms of layout and structure. It explains and analyzes the method selection, model establishment, model application, test comparison and test condition setting of river bank slope stability analysis. The analysis of the influencing factors of stability is also relatively comprehensive, and fully considered variable control experiments and model stability analysis have been carried out. The selection of the article from a novel perspective and the writing of the article are quite bright.

But in the article, there are also some small problems as follows:

1. It is suggested that the abstract part should focus on the effect of freezing and thawing, because the title of the article is the effect of freezing and thawing on the stability of river banks. That is to say, the main influencing factor of this part is freeze-thaw, and osmotic pressure is a factor that affects the stability of river bank induced by freeze-thaw.

2. There are differences in the description of the content of the abstract. According to the content of the article, it should be the effect of freezing and thawing on the internal friction angle and cohesion of the four types of water content samples, but the content of the abstract appears ''degree of saturation'', which leads to this problem. A sentence would be understood to cause desaturation.

3. In the process of analyzing the changes in the stability of the bank slope caused by the rise and fall of the water level, it is recommended to combine it with the actual situation. For example, in the process of the rise of the river water level, on the one hand, it will form a height difference with the groundwater level of the river bank, which will lead to the sliding surface. On the other hand, the seepage caused by the rise of the river water level will affect the water content of the bank slope soil, and the change of the water content will affect the size of C and φ. Therefore, the impact caused by the rise of river water level should be analyzed in many aspects.

4. The sources and test method recommendations for effective cohesion and effective internal friction angle are explained in table 3, which can increase verifiability and repeatability. Secondly, although freezing and thawing will cause damage to the soil structure and affect its mechanical properties, whether this effect will be eliminated or there should be experimental proof of the elimination (because the effective internal friction and effective cohesion during the flood period in Table 3 are both On the other hand, the increase in soil moisture content during the flood period will definitely lead to a decrease in the effective cohesion and internal friction angle of the soil body, but it is increased in Table 3, and there should be relevant analysis and explanation .

5. During the freeze-thaw cycle test, the content of the article should be a freeze-thaw cycle test with remodeled soil, and it is recommended to explain. Moreover, taking the actual soil moisture content of 21% as the control moisture content, it should be stated that 21% is the moisture content of the original soil in which period, such as the freezing period, the melting period, or the flood period. Because moisture content has a significant effect on freezing and thawing, it should be explained.

Author Response

July. 22 2022

Thank you for your letter and valuable comments on our manuscript entitled “Freeze-Thaw Effect on River Bank Stability” (Manuscript ID. water-1822936).  We have revised the paper based on review comments.

If you have any additional comments and/or concerns, please do not hesitate to contact us directly.

Sincerely,

 Chao Li, Zhen Yang *, Hung Tao Shen, and Xianyou Mou

Response to review comments

Response to comments of Editors

Point 1 : Please check that all references are relevant to the contents of the manuscript.

 Response: Thanks for the suggestions. We have already check all references, and those references are relevant to the contents of the manuscript.

Point 2: Any revisions to the manuscript should be marked up using the “Track Changes” function if you are using MS Word/LaTeX, such that any changes can be easily viewed by the editors and reviewers.

Response: Thanks for the suggestions. The revisions to the manuscript have already marked up using the “Track Changes” function. At the same time, we also summit a clearn version.

  Response to review comments

 Response to comments of Reviewer 1

Point 1: In general, this article is reasonable in terms of layout and structure. It explains and analyzes the method selection, model establishment, model application, test comparison and test condition setting of river bank slope stability analysis. The analysis of the influencing factors of stability is also relatively comprehensive, and fully considered variable control experiments and model stability analysis have been carried out. The selection of the article from a novel perspective and the writing of the article are quite bright.

 Response: We appreciate your positive comments. We made additional improvements on the manuscripts as suggested.  

But in the article, there are also some small problems as follows:

Point 2: Comment: It is suggested that the abstract part should focus on the effect of freezing and thawing, because the title of the article is the effect of freezing and thawing on the stability of river banks. That is to say, the main influencing factor of this part is freeze-thaw, and osmotic pressure is a factor that affects the stability of river bank induced by freeze-thaw.
Response: Thank you for your constructive suggestion. We revised the abstract based on the suggestion.

Point 3: Comment: There are differences in the description of the content of the abstract. According to the content of the article, it should be the effect of freezing and thawing on the internal friction angle and cohesion of the four types of water content samples, but the content of the abstract appears ''degree of saturation'', which leads to this problem. A sentence would be understood to cause desaturation.

Response: Yes, we corrected it in the revised abstract.

Point 4: Commnet: In the process of analyzing the changes in the stability of the bank slope caused by the rise and fall of the water level, it is recommended to combine it with the actual situation. For example, in the process of the rise of the river water level, on the one hand, it will form a height difference with the groundwater level of the river bank, which will lead to the sliding surface. On the other hand, the seepage caused by the rise of the river water level will affect the water content of the bank slope soil, and the change of the water content will affect the size of C and φ. Therefore, the impact caused by the rise of river water level should be analyzed in many aspects.

 Response: Thank you for the comment. During the river freeze period, the presence of river ice cover causes the river water level to rise, which in turn causes changes in the water content of the bank material. Similarly, during the thawing period, due to the melting of river ice cover and the release of water from the channel storage, the water level of the river will drop. This causes further adjustment of the moisture content of the bank material. The changes in the moisture content of the bank material will cause changes in the cohesion and friction angle, which in turn impact the stability of the bank. Therefore, the stability of the bank under the interaction of different river water levels and groundwater levels is discussed in the paper based on the changes of river water level, which leads to the conclusion that for a given river water level, a higher groundwater level will result in a smaller safety factor of bank stability. Similarly, for a given groundwater level, a lower the river water level will lead to a smaller safety factor. We modified the text before Figure 6.

Point 5: Comment: The sources and test method recommendations for effective cohesion and effective internal friction angle are explained in table 3, which can increase verifiability and repeatability. Secondly, although freezing and thawing will cause damage to the soil structure and affect its mechanical properties, whether this effect will be eliminated or there should be experimental proof of the elimination (because the effective internal friction and effective cohesion during the flood period in Table 3 are both On the other hand, the increase in soil moisture content during the flood period will definitely lead to a decrease in the effective cohesion and internal friction angle of the soil body, but it is increased in Table 3, and there should be relevant analysis and explanation.

Response: The thawing period and the subsequent dry period are considered to be the periods when the soil has experienced the freeze-thawing effect, so the cohesion and internal friction angle of the soil in these two periods are calculated using the data after freeze-thawing. In other periods (including water level rise period, flood period and receding water period), the soil does not experience the freeze-thaw process, and the physical and mechanical properties of the soil are the values without considering the effect of freeze-thaw, and the cohesion and internal friction angle of the soil without the freeze-thaw effect are used in the calculation. Therefore, the cohesive force of the soil in the calculation of slope stability in the thawing and dry water periods is smaller than that in the rising water level, flooding and receding water periods.

 Point 6: Comment: During the freeze-thaw cycle test, the content of the article should be a freeze-thaw cycle test with remodeled soil, and it is recommended to explain. Moreover, taking the actual soil moisture content of 21% as the control moisture content, it should be stated that 21% is the moisture content of the original soil in which period, such as the freezing period, the melting period, or the flood period. Because moisture content has a significant effect on freezing and thawing, it should be explained.

 Response: The triaxial test samples in this study are remodeling soil samples. The samples were collected on July 15, 2020, during the high water period, at Shishifenzi upstream of the Toudaoguai hydrological station. The natural water content of the sampled soil was 21%.  Hence, the cohesion and internal friction angle values with a soil water content of 21% were used to calculate the bank safety factor when analyzing bank slope stability. The stability of bank slopes under other water contents was not carried out, and only the effects of freeze-thaw action on soil cohesion and internal friction angle under different water contents were analyzed. We modified the paragraph after Figure 8.

Reviewer 2 Report

Abstract

·      You should mention the place where the study was carried out: Shisifenzi Bend

·      This information is not clear! Please rewrite and explain better what you want to convey. How many soil moisture contents did you analyze? If were 4, the information is more noticeable!

“Freeze-thaw action affects the cohesiveness of bank soil samples with moisture contents of 15%, 18%, and 21%, and the degree of saturation decreased by 34.37 %, 30.71%, 32.48%, and 46.23%, respectively. In addition, the internal friction angle is reduced by 1.78%, 2.74%, 6.33%, and 5.32%, resulting in a 24.35% to 29.13% reduction in the safety factor of bank stability”.

Introduction

·      You should better explain the objectives of the work

·      It always made reference to "Inner Mongolia section of the Yellow River" and suddenly "Shisifenzi Bend in the Yellow River" appears. You have to contextualize geographically this last reference relative to the first!

3.1. Study area

·      If possible, place a map with the geographic location of the study área

·      Make a brief climatic characterization of the study area (temperature, precipitation, ...)

·      Table 1 - Put the meaning of the abbreviations in the table caption

·      Figure 7 - This Figure shows sediment data from the Shizuishan and Toudaoguai hydrological stations. What is the relationship between these hydrological stations and the study area?

Overall:

The structure of the article is a bit confusing, with a mix between methods, results, and discussion!

The results are interesting and deserve a better presentation and discussion so that the information is clearer.

Author Response

Response to comments of Reviewer 2

Overall :

Point 1: The structure of the article is a bit confusing, with a mix between methods, results, and discussion!

The results are interesting and deserve a better presentation and discussion so that the information is clearer.

Response: Thanks for the suggestions. We revised the paper to improve it presentation and discussion.

 Other comments:

Point 2: Comment: You should mention the place where the study was carried out: Shisifenzi Bend

Response: Agree. We modified the text and specified the location, and added a new Figure 1 showing its location.

Point 3: Comment: This information is not clear! Please rewrite and explain better what you want to convey. How many soil moisture contents did you analyze? If were 4, the information is more noticeable!

“Freeze-thaw action affects the cohesiveness of bank soil samples with moisture contents of 15%, 18%, and 21%, and the degree of saturation decreased by 34.37 %, 30.71%, 32.48%, and 46.23%, respectively. In addition, the internal friction angle is reduced by 1.78%, 2.74%, 6.33%, and 5.32%, resulting in a 24.35% to 29.13% reduction in the safety factor of bank

Response: Yes, there were 4. We corrected this. Thanks for catching this error.

 Point 4: Comment: You should better explain the objectives of the work.

Response: Thank you for your suggestion. We modified the manuscript to make it clearer.

Point 5: Comment: It always made reference to "Inner Mongolia section of the Yellow River" and suddenly "Shisifenzi Bend in the Yellow River" appears. You have to contextualize geographically this last reference relative to the first!

Response: Thank you for the comment. We add a figure (see Comment 5) to show the river reach and the study sites geographically with land marks. We also improved the text to make these clearer. 

 Point 6: Comment: If possible, place a map with the geographic location of the study área.

Response: The following map is added as Figure 1 in the manuscript:

Figure 1. Inner Mongolia reach of the Yellow River, and the Shisifenzi Bend section.

Point 7: Comment: Table 1 - Put the meaning of the abbreviations in the table caption

Response: In Table I, c and c' are the cohesive force and effective cohesion force, Ï• and Ï•' are the internal friction angle and effective internal friction angle. Their meanings are explained in the preceding section. They are abbreviated in the table because the parameter names are too long and difficult to include in the table.

 Point 8: Comment: Figure 7 - This Figure shows sediment data from the Shizuishan and Toudaoguai hydrological stations. What is the relationship between these hydrological stations and the study area?

Response: These information are now added in the text, together with the new figure 1 shown in Comment 5 above. 

Reviewer 3 Report

It is a very interesting work on this type of rivers affected by these two clear freeze-thaw stages that allows the application to this type of scenarios in China and by extension in other parts of the world equally affected.

The introduction is very correct in presenting the problem. Section 2 presents the calculation diagrams and the forces involved in the river bank slope, as well as the diagram of the two water contact zones, the river and its shore and the level of the groundwater layer of the slope.

The main objective of this work is to analyse the bank stability and to facilitate the calculation according to the flat slip analysis following the initial model of Osman and Thorne and modified by Darby and Thorne as it did not take into account the lateral water pressure. 

Figures 2 and 3 are taken exactly from Darby and Thorne's publication and are not cited in each figure (i.e. after Darby and Thorne).

The results obtained in the different selected periods with respect to the freeze-thaw periods and therefore the increase or decrease of water levels in the river and the groundwater layer have enabled a series of conclusions to be drawn which allow safety measures to be implemented in this type of case.

The fourth and last conclusion could be omitted and there would be no major problem.

Author Response

Response to comments of Reviewer 3

 We appreciate the positive and constructive comments. Following are our responses to specific comments:

Point 1: Comment: Figures 2 and 3 are taken exactly from Darby and Thorne's publication and are not cited in each figure (i.e. after Darby and Thorne).

Response: Figure 3 is modified from the figure in Darby and Throne (Ref. 26). The figure in Ref. 26 does not consider osmotic water pressure, which is considered in Figure 3. The following figure marked the changes from that of Darby and Throne.

Point 2: Comment: The results obtained in the different selected periods with respect to the freeze-thaw periods and therefore the increase or decrease of water levels …. Allow safety measures to be implemented in this type of case. The fourth and last conclusion could be omitted and there would be no major problem.

Response: Thank you for your comment. The bank slope angle is an important factor affecting the stability of the embankment. By analyzing the stability of the bank for different bank slopes, the stable bank slope angle at different water level conditions and soil structure is determined. This could guide the design slope angle in bank protection projects. Hence, we would like to retain this conclusion with modification.
